# Parameters Optimization of Auxiliary Gas Process for Double-Wire SS316L Stainless Steel Arc Additive Manufacturing

**Wei Wu [1,2], Jiaxiang Xue [3], Wei Xu [1,2], Hongyan Lin [4], Heqing Tang [4] and Ping Yao [4,\*]**

1    School of Automobile and Transportation Engineering, Guangdong Polytechnic Normal University, Guangzhou 510450, China; wuwei_5v@gpnu.edu.cn (W.W.); xuy_wei@126.com (W.X.)
2    Guangzhou Key Laboratory of Thermal Safety Technology for New Energy Vehicle Power System, Guangzhou 510450, China
3    School of Mechanical and Automotive Engineering, South China University of Technology, Guangzhou 510640, China; mejiaxue@scut.edu.cn
4    School of Electrical and Mechanical, Guangdong Polytechnic Normal University, Guangzhou 510635, China; sjy843644862@163.com (H.L.); happythq2015@163.com (H.T.)
\*    Correspondence: gsyaop@gpnu.edu.cn; Tel.: +86-020-3825-6601

**Abstract:** Serious heat accumulation limits the further efficiency and application in additive manufacturing (AM). This study accordingly proposed a double-wire SS316L stainless steel arc AM with a two-direction auxiliary gas process to research the effect of three parameters, such as auxiliary gas nozzle angle, auxiliary gas flow rate and nozzle-to-substrate distance on depositions, then based on the Box–Behnken Design response surface, a regression equation between three parameters and the total score were established to optimized parameters by an evaluation system. The results showed that samples with nozzle angle of 30° had poor morphology but good properties, and increasing gas flow or decreasing distance would enhance the airflow strength and stiffness, then strongly stir the molten pool and resist the interference. Then a diverse combination of auxiliary process parameters had different influences on the morphology and properties, and an interactive effect on the comprehensive score. Ultimately the optimal auxiliary gas process parameters were 17.4°, 25 L/min and 10.44 mm, which not only bettered the morphology, but refined the grains and improved the properties due to the stirring and cooling effect of the auxiliary gas, which provides a feasible way for quality and efficiency improvements in arc additive manufacturing.

**Keywords:** double-wire; arc additive manufacturing; assessment; auxiliary gas; optimization

## 1. Introduction

Wire arc additive manufacturing (WAAM) has been extensively used in manufacturing diverse metal materials. It has advantages including a high deposition rate, high material utilization, and relatively high production and low equipment costs, which was stated by Martina et al. [1]. Double-wire arc additive manufacturing can address the conflict between improving deposition rate and reducing heat input. In the study of double-wire cold metal transfer plus pulse (CMT + P) additive manufacturing researched by Wu et al. [2], increasing wire feeding speed and scanning speed resulted in serious heat accumulation, which decreased the height and increased the width of the specimens. Although the heat input keeps constant, the severe thermal accumulation would lead to the coarse grain and poor performance; therefore, it is difficult to obtain all the objects optimization, such as deposition efficiency, morphology and performance by adjusting parameters. Accordingly, several additional auxiliary processes have been carried out to reduce cooling rates and improve performance by researchers.

Coregrove et al. [3] and Hönnige et al. [4] developed a cross-track cold rolling system that pressed each layer with a roller after depositing, which could not only reduce deformation and surface roughness, but also refine the microstructure and improve the longitudinal

tensile strength. In addition, Mcandrew et al. [5] adopted a new "reversed roll" to refine the grains by using plasma arc tungsten inert gas welding to fabricate the Ti-6Al-4V part. Moreover, Gu et al. [6,7] studied the changing size and formation mechanism of the sample porosity after rolling treatment for aluminum alloy CMT additive manufacturing. The results showed that rolling could significantly reduce the porosity number, and the larger the rolling load, the smaller the diameter of the porosity; therefore, the hardness, strength and elongation of the deposited samples after rolling were increased by 40%, 18.2% and 20%, respectively. However, due to the geometric limitations of the rolling process, complex and expensive motion systems need to be considered to achieve effective rolling processes for complex molding. Additionally, the wall after rolling has a smaller height and bigger width, which limited its industrial applications. On the other hand, Martina et al. [8] and Williams et al. [9] referred that the rolling temperature might be required much lower than the deposition interlayer temperature, which would increase the total time of AM and greatly reduce the efficiency.

Wang et al. [10] deposited H13 steel using metal inert gas welding, and the tensile properties of the samples were isotropic after annealing at 830 °C for 4 h. Then, Chen et al. [11] analyzed the microstructure evolution and mechanical properties of 316L AM parts after heat treatment, and the results showed that the heat treatment temperature made the σ phase disappear, thus reducing the possibility of crack generation and increased corrosion resistance. Additionally, Caballero et al. [12] obtained the required tensile properties of 17-4 PH stainless steel and Qi et al. [13] obtained double-wire 2024 alloy parts by WAAM and processed them by heat treatment. The above findings indicate that the post-heat treatment process can change the microstructure and improve the performance, but cannot improve the molding morphology. Furthermore, the post-heat treatment needs to be carried out separately after AM, which is also time-consuming and expensive.

Liu [14] applied the longitudinal magnetic field to aluminum alloy AM research, and the results showed that the uneven surface of the specimens after adding a magnetic field was improved, as well as the mechanical properties. In addition, Zhou et al. [15] used a magnetic field to fabricate low-carbon alloy steel parts by gas metal arc welding (GMAW), which effectively improved the lapping accuracy and the surface quality. In addition, the electromagnetic stirring force could also refine the grains and reduce the molding defects, and decrease the anisotropy of mechanical properties. The greater the magnetic field strength, the better the grain refinement; however, this is in equilibrium with the increase of surface roughness and burnout, which limits the production of high quality WAAM parts. Moreover, the applied magnetic field needs a complex device and is difficult to control.

Wu et al. [16] conducted a WAAM system of Ti6Al4V thin-walled structure with forced cooling and compressed $CO_2$. This cooling gas could reduce surface oxidation and microstructure size, and enhance the hardness and strength. In addition, manufacturing efficiency was considerably improved due to the decreasing time between layers. However, cooling gas in the whole process only played a cooling role on the depositing part, instead of stirring effect on the liquid molten pool; therefore, the performance improvement was not apparent.

Li et al. [17] designed a thermoelectric cooling device for manufacturing 2325 aluminum alloy thin-walled structures by GMAW. The findings indicated that the device could maintain stable heat dissipation and decrease the interlayer waiting time by 60.9%, and additionally improve the morphology and mechanical properties. However, the device is quite complex and costly for extensive application.

Based on the above studies, a two-direction auxiliary gas platform was established for multi-layer deposition of double-arc additive manufacturing, which could stir and cool the molten pool by the gas jet during the deposition process, and consequently improve the molding and performance. Specifically, three processes of the auxiliary nozzle angle, the auxiliary gas flow rate and the auxiliary gas nozzle-to-substrate distance were studied for different impact force and gas jet range. Additionally, the response surface method is an experimental design and analysis method that uses mathematical and statistical methods to

model, analyze and optimize multi factor response problems through reasonably designed experiments, so as to finally achieve the optimization of objectives. The Box–Behnken Design (BBD) is a form of response surface design, which does not include embedding factor or partial factor design. BBD has a processing combination located at the midpoint of the edge in the experimental space, and requires at least three factors. The authors of [18,19] applied this to establish a regression model between the auxiliary process parameters, forming and performance to optimize the parameters.

## 2. Materials and Methods

### 2.1. Test Materials and Equipment

Two SS316L stainless-steel welding wires with 1.2 mm diameter and a 250 mm × 100 mm × 5 mm base plate were selected. Before the test, the substrate surface was polished. The double-wire arc AM platform consisted of a VR7000 CMT wire feeder (Fronius International GmbH, Wels, Austria), shielding gas, auxiliary gas device, industrial control computer and high precision KUKA six-axis robot (KUKA Roboter GmbH, Augsburg, Germany). As shown in Figure 1, the auxiliary gas device is mainly composed of an argon cylinder, pressure reducing valve, airflow splitter, and flow control valve and the gas nozzle. The auxiliary gas flow is mainly used to stir the unsolidified molten pool and protect the deposited surface from air oxidation.

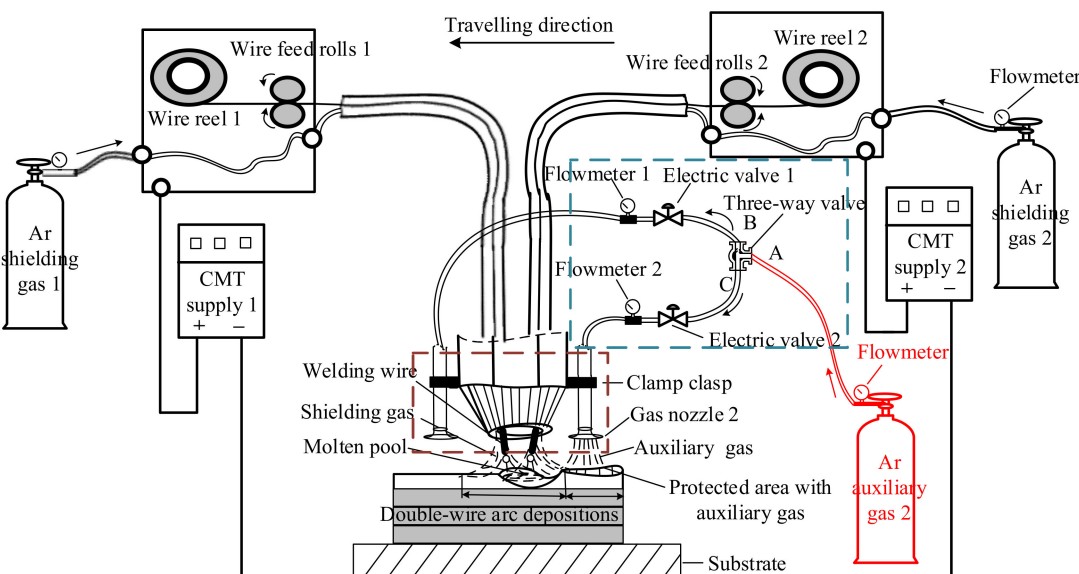

**Figure 1.** Platform of auxiliary gas process in double-wire SS316L stainless steel arc additive manufacturing.

Each welding gun was clamped with an auxiliary gas nozzle, which alternately realized the auxiliary gas inject behind the welding torch in two directions, as shown in Figure 1. The auxiliary free jet impacted the current pool, then the jet force carried with kinetic energy could affect the dynamic equilibrium of the molten pool to produce deformation and play a role in the stirring pool and reducing surface oxidation, which could significantly improve the deposition forming and performance.

### 2.2. Evaluation Method

Different evaluation indexes, such as surface roughness, morphology efficiency hardness, tensile strength and elongation have different units and intervals. In order to achieve indicator comparability and avoid the impact of diverse ranges on results, feature data should be normalized before evaluation. The normalization is a process of converting the actual value to the evaluation value, which is also called the dimensionless index. Then,

the linear normalization method is calculated according to Formulas (1) and (2) to achieve normalized processing.

$$f(x) = \frac{(x - x_{min})}{(x_{max} - x_{min})} \tag{1}$$

$$f(x) = \frac{(x_{max} - x)}{(x_{max} - x_{min})} \tag{2}$$

where $x_{min}$ and $x_{max}$ represent the minimum and maximum value of $x$, respectively. If the index is positive, that is, the bigger the value, the better the evaluation, and then Formula (1) should be selected. While the index is negative, that is, the bigger the index value, the worse the evaluation, such as the molding defect and the roughness, then the negative normalization should be used, as shown in Formula (2), and the maximum and minimum value can be obtained from the test data. Consequently, being dependent on this method, the data are uniformly classified between 0 and 1.

Figure 2 shows the schematic diagram of calculating the side surface roughness by cross-section, where, $d_1$ and $d_3$ indicate the width of the wave troughs, $d_2$ and $d_4$ indicate the width of the wave peaks, and thus the width difference of the wave peak and trough at each layer can be approximately regarded as the AM surface roughness, as shown in Formula (3).

$$S_a = \frac{\sum_1^{30} (d_{i+1} - d_i)}{N \times 2} \tag{3}$$

where, $S_a$ represents the side surface roughness, $d_i$ represents the width of wave peaks and troughs at each layer, and $N$ represents the deposited layer number.

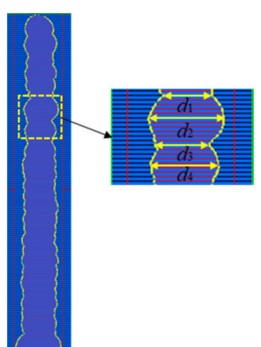

**Figure 2.** Schematic illustration of calculating surface roughness on the cross-section.

The roughness along sample height or width was obtained from the difference between the maximum height or width of each section, as shown in Formula (4).

$$S_b = \frac{\sum_1^{n-1} (D_{i+1} - D_i)}{n - 1} \tag{4}$$

where $S_b$ indicates the surface roughness along sample height or width direction, $D_i$ indicates the maximum height or width value of each section, and $n$ indicates the section number.

Based on the above formulas, the ultimate roughness was determined by the average roughness of three directions, which was one of the evaluation indexes of molding quality. The morphology efficiency was obtained depending on the calculation method reported by Wu et al. [20].

### 2.3. Experiment Conditions

As the jet impact force is associated with the gas flow and the nozzle position, the position relationship between the auxiliary gas nozzle and the welding torch is shown in Figure 3, where $H_{AG}$ and $\theta_{AG}$ represent the nozzle-to-substrate distance, and the angle

between the nozzle and welding torch, respectively. Owing to the millisecond of the liquid stainless steel solidification, the big distance would make the auxiliary gas unable to stir the liquid molten pool. Taking the wire extension of 12 mm and the arc length of 2 mm as the standard, 4, 9 and 14 mm were chosen for the nozzle-to-substrate distance. Additionally, the pre-test indicated that when the angle between the auxiliary gas nozzle and welding torch was negative, the auxiliary gas jet would be kept away from the liquid molten pool, which would only cool the deposited layer weakly and rather than stir the liquid molten pool. However, when the angle exceeded 30°, the arc adjacent to the auxiliary nozzle was obviously blown off, which resulted in an unstable deposition process. Consequently, three angles of 0°, 15° and 30° were selected for this study.

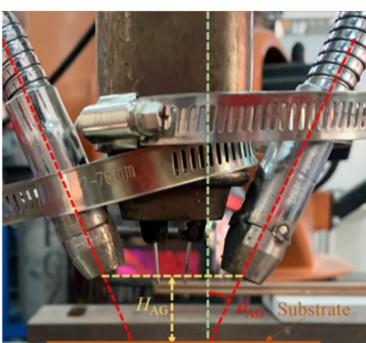

**Figure 3.** Position diagram of auxiliary gas nozzle and welding gun.

CMT + P mode was used for two-direction deposition, and the deposition current, voltage, wire feeding speed and scanning speed were 103 A, 16.8 V, 5 m/min and 1.2 m/min, respectively. Then, morphology and performance of deposition were assessed by a quantitative evaluation system. After that the model between the total score and three process parameters was established and checked, then the influence of each process parameter on the forming, microstructure and properties of deposition was discussed. Finally, optimal auxiliary gas process parameters were obtained according to the model prediction.

The most central stable part near the vertical wall body was selected, and a transverse section with 10 mm width was cut to prepare microstructural and hardness sample. The greasy dirt on the surface of each metallographic sample was wiped away using acetone, and the sample was inlaid through a cold inlay process. First, 180#, 600#, and 800# sandpapers were used for rough grinding. Then, 1200#, 2500#, and 5000# sandpapers were used for fine grinding, followed by polishing with a 2.5 μm particle diamond polishing agent. Finally, the metallographic samples were corroded using aqua regia ($HCl:HNO_3 = 3:1$) for 30 s, before being rinsed and dried with water and alcohol, respectively. The microstructural characteristics of the depositions were analyzed with an optical metallographic microscope. Then microhardness along the deposited direction was measured by a HMV-2T micro Vickers hardness tester with a load of 0.5 kg maintaining for 10 s during the test. In addition, four horizontal stretch samples were extracted from the deposition, in accordance with the GB/T 228-2002 standard, tensile tests were conducted parallel to the welding scanning direction at a speed of 2 mm/min at room temperature by an electronic universal testing machine CMT5105 with a maximum load of 100 kN. A NOVA NANO scanning electron microscope (SEM) 430 (FEI, Eindhoven, The Netherlands) was utilized to observe the fracture morphology of snapped samples.

## 3. Results

The morphology and performance results of the BBD experiment are shown in Table 1. Based on the BBD response surface method, this study designed 17 samples, which had five repeated samples as 1#, 3#, 7#, 12# and 17#; therefore, sample 1# was the representation of analysis.

**Table 1.** Morphologies and properties by BBD response surface experiment (BBD represents Box–Behnken Design).

| Samples | Shape Appearance and Central Cross-Section Morphology | Average Hardness/HV | Average Tensile Strength/MPa | Average Elongation/% |
|---|---|---|---|---|
| 1# |  | 180 ± 4.53 | 499.87 ± 2.91 | 49.23 ± 2.26 |
| 2# |  | 190.35 ± 5.2 | 512.88 ± 7.37 | 48.07 ± 2.06 |
| 4# |  | 186.13 ± 4.57 | 510.47 ± 2.29 | 48.17 ± 1.92 |
| 5# |  | 182.75 ± 5.53 | 503.55 ± 1.63 | 47.24 ± 0.3 |
| 6# |  | 176.1 ± 3.03 | 490.77 ± 3.57 | 50.65 ± 2.18 |
| 8# |  | 183.25 ± 5.44 | 507.41 ± 6.57 | 47.3 ± 0.68 |
| 9# |  | 184.6 ± 3.88 | 504.54 ± 2.9 | 50.86 ± 1.97 |
| 10# |  | 176.42 ± 4.75 | 494.51 ± 4.86 | 50.31 ± 3.1 |
| 11# |  | 174.7±1.97 | 470.11 ± 13.37 | 52.76 ± 3.1 |
| 13# |  | 175.05 ± 3.87 | 494.93 ± 2.3 | 50.79 ± 2.2 |
| 14# |  | 173.43 ± 3.46 | 483.25 ± 6.24 | 49.73 ± 3.11 |
| 15# |  | 172.53 ± 3.16 | 478.34 ± 12.3 | 50.7 ± 1.42 |
| 16# |  | 176.36 ± 2.65 | 481.43 ± 4.68 | 51.43 ± 3.45 |

The auxiliary gas process parameters of 17 group experiments designed by the BBD-based surface response method are presented in Table 2, the morphology defect, roughness and molding efficiency were evaluated by the quantitative evaluation system, and the molding scores were obtained, as shown in Table 2.

**Table 2.** Experimental groups of the BBD response surface methodology (BBD represents Box–Behnken Design).

| Samples | Angle between Nozzle and Welding Torch $\theta_{AG}/°$ | Gas-Flow Rate $L_{AG}/L·min^{-1}$ | Distance between Nozzle and Substrate $H_{AG}/mm$ | Molding Score | Performance Score | Comprehensive Score |
|---|---|---|---|---|---|---|
| 1# | 15 | 20 | 9 | 79.88 | 48 | 63.94 |
| 2# | 30 | 25 | 9 | 40.32 | 66 | 53.16 |
| 3# | 15 | 20 | 9 | 78.58 | 49.58 | 64.08 |
| 4# | 30 | 15 | 9 | 61.47 | 57.94 | 59.7 |
| 5# | 15 | 25 | 4 | 67.83 | 40.66 | 54.25 |
| 6# | 0 | 20 | 4 | 45.22 | 45.21 | 45.21 |
| 7# | 15 | 20 | 9 | 78.35 | 48.84 | 63.6 |
| 8# | 30 | 20 | 4 | 56.28 | 44.65 | 50.46 |
| 9# | 30 | 20 | 14 | 31.67 | 70.7 | 51.19 |
| 10# | 15 | 15 | 4 | 87.50 | 45.91 | 66.7 |
| 11# | 0 | 15 | 9 | 59.95 | 43.65 | 51.8 |
| 12# | 15 | 20 | 9 | 77.78 | 49.2 | 63.49 |
| 13# | 15 | 25 | 14 | 74.59 | 47.2 | 60.89 |
| 14# | 15 | 15 | 14 | 62.73 | 28.8 | 45.77 |
| 15# | 0 | 20 | 14 | 44.68 | 30.7 | 37.69 |
| 16# | 0 | 25 | 9 | 45.05 | 45.4 | 45.23 |
| 17# | 15 | 20 | 9 | 78.32 | 47.87 | 63.1 |

### 3.1. Morphology Evaluation

Compared with other angles, it can be seen from Table 1 that samples with the auxiliary nozzle angle of 30° produced defects easier, which was mainly due to the fact that the jet was strongly pushing the molten pool along the walking direction, that is, pushing the molten pool forward. The nozzle angle of 0° caused the specimen ends to be large, which was owing to the fact that the auxiliary gas jet had little stirring and cooling effect on the molten pool, then lead to the serious accumulation of the end part. The smaller the angle, the further horizontal distance of the auxiliary gas jet from the welding wire, that is, the gas jet from the liquid molten pool was too far to stir the pool, and with the increasing number of the layer, the jet pressure decreased and the protection range increased. It can be seen from the cross-section morphology that the width and the roughness became bigger at 30°, while specimens of 0° were obviously thinner and higher, and a narrower necking position appeared at the bottom, which was mainly due to the little stirring effect on the molten pool and the rapid cooling of the first two layers, which consequently increased the roughness and reduced the effective deposition area. However, the morphology of 15° was more uniform, then the roughness was the smallest and thus the forming efficiency was improved.

### 3.2. Microstructure of Depositions

As shown in Figure 4, especially in the zoom image of sample #11, the microstructure mainly presents columnar crystals, which was fundamentally composed of austenite and ferrite. However, due to the diversity of the auxiliary gas process parameters, the crystal size of samples was different. Compared with the auxiliary nozzle angle of 0°, the columnar crystals length and the secondary dendrite arm distance were reduced with angle 15° and 30°, which further indicated that the auxiliary gas had a stirring and cooling effect on the unsolidified molten pool.

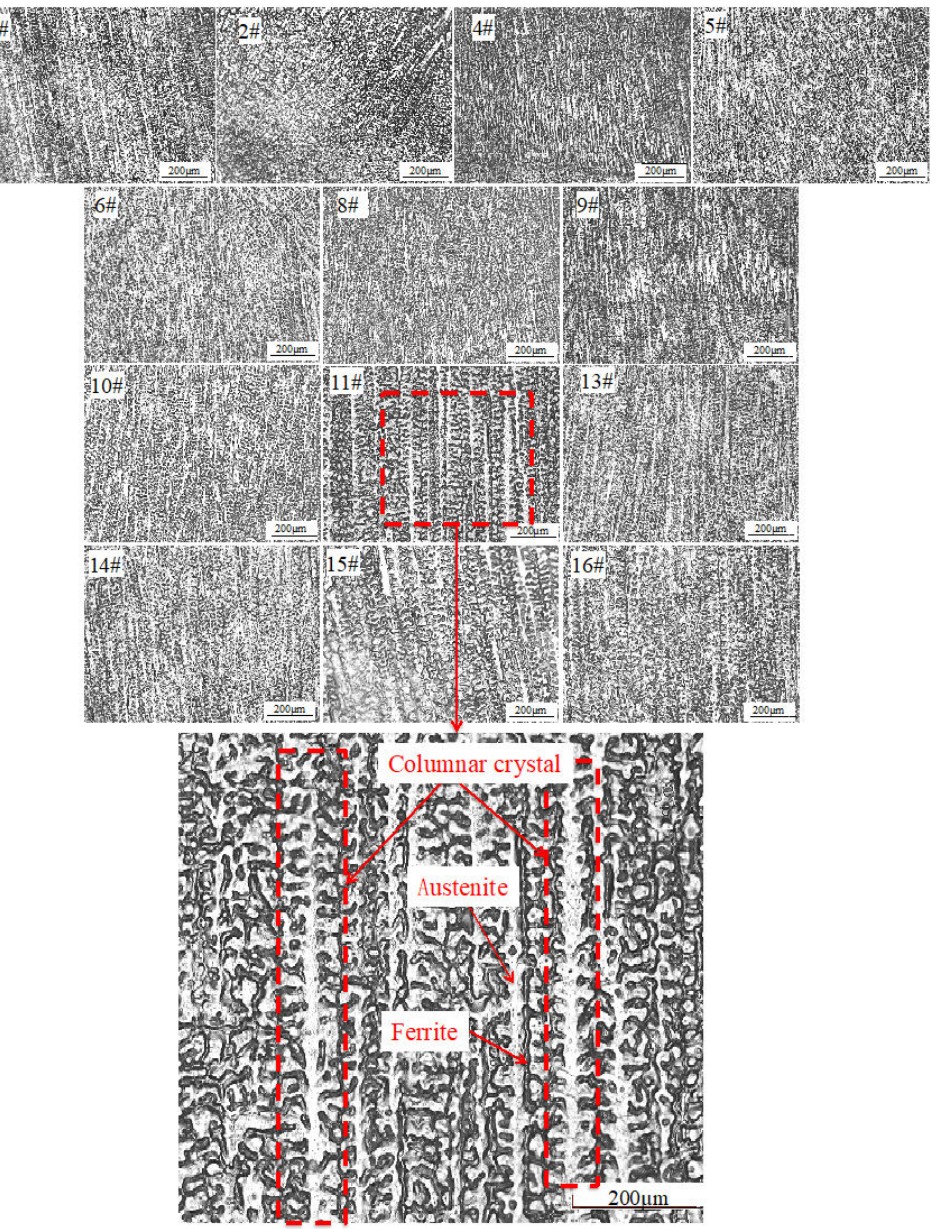

**Figure 4.** Microstructure of different samples.

*3.3. Microhardness*

The Vickers hardness test was carried out on 17 groups of specimens, and the average hardness was obtained, as shown in Figure 5 and Table 1, and it indicated that the hardness of 2# with the auxiliary torch angle of 30°, flow of 25 L/min and distance of 9 mm was the highest, followed by 4# with angle of 30°, flow of 15 L/min and distance of 9 mm, while the hardness of 5# with angle of 15°, flow of 25 L/min and distance of 4 mm, 8# with angle of 30°, flow of 20 L/min and distance of 4 mm, 9# with angle of 30°, flow of 20 L/min and distance of 14 mm, and 1# with angle of 15°, flow of 20 L/min and distance of 9 mm was slightly higher, which directly corresponded to the microstructure size. The different combination of auxiliary gas process parameters affected the cooling rate and the stirring degree, which would further affect the microstructure and the properties.

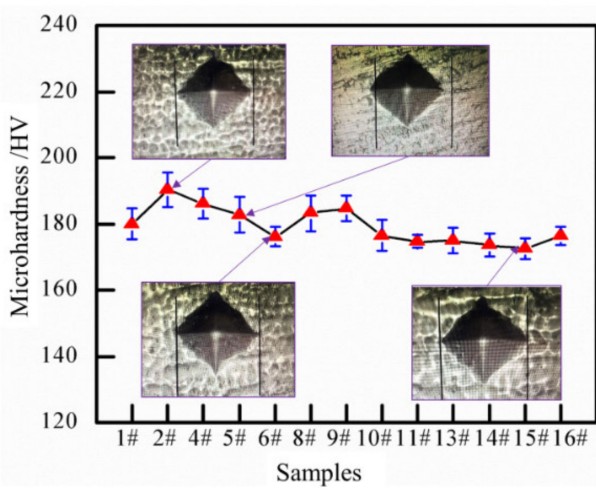

**Figure 5.** Microhardness of samples 1#–17#.

*3.4. Tensile Properties*

The stress–strain curves measured at the same position of tensile specimens are illustrated in Figure 6, and the average tensile strength and elongation are provided in Table 1. Compared with the sample without auxiliary gas process manufactured by Wu et al. [2], both the tensile strength and elongation were increased, which indicated that the auxiliary gas process could improve the performance. In addition, according to the average tensile strength, it could be seen that the auxiliary gas of 30° had some stirring and cooling effect on the unsolidified molten pool, resulting in a relatively better tensile strength performance. However, when the nozzle was 0°, the airflow was far away from the unsolidified molten pool, which only played a cooling role and made the performance relatively poor. Furthermore, increasing the auxiliary gas flow or decreasing the distance would also enhance the airflow strength and stiffness, and thus improve the ability to stir the molten pool and resist the interference; despite all this, samples of 30° with increased strength decreased the elongation.

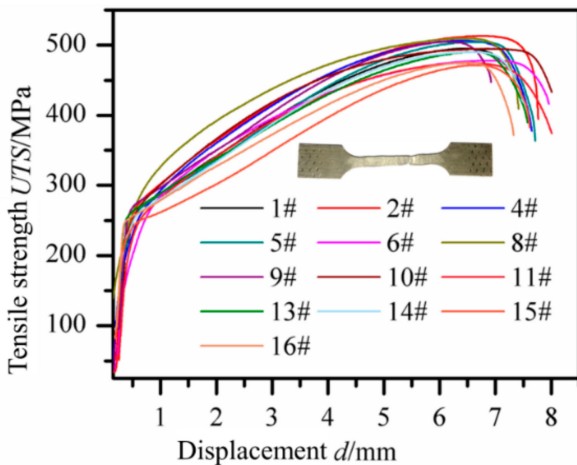

**Figure 6.** Tensile results of specimens.

The fracture surface morphology presented a large number of dimples after the tensile test, as shown in Figure 7, which indicated that all the specimens were typical ductile fractures. However, 6#, 9#, 10#, 11#, 15# and 16# showed larger and deeper equiaxial dimples with obvious tearing edges compared to other specimens, which stated that the plasticity of these specimens was preferable to that of other specimens.

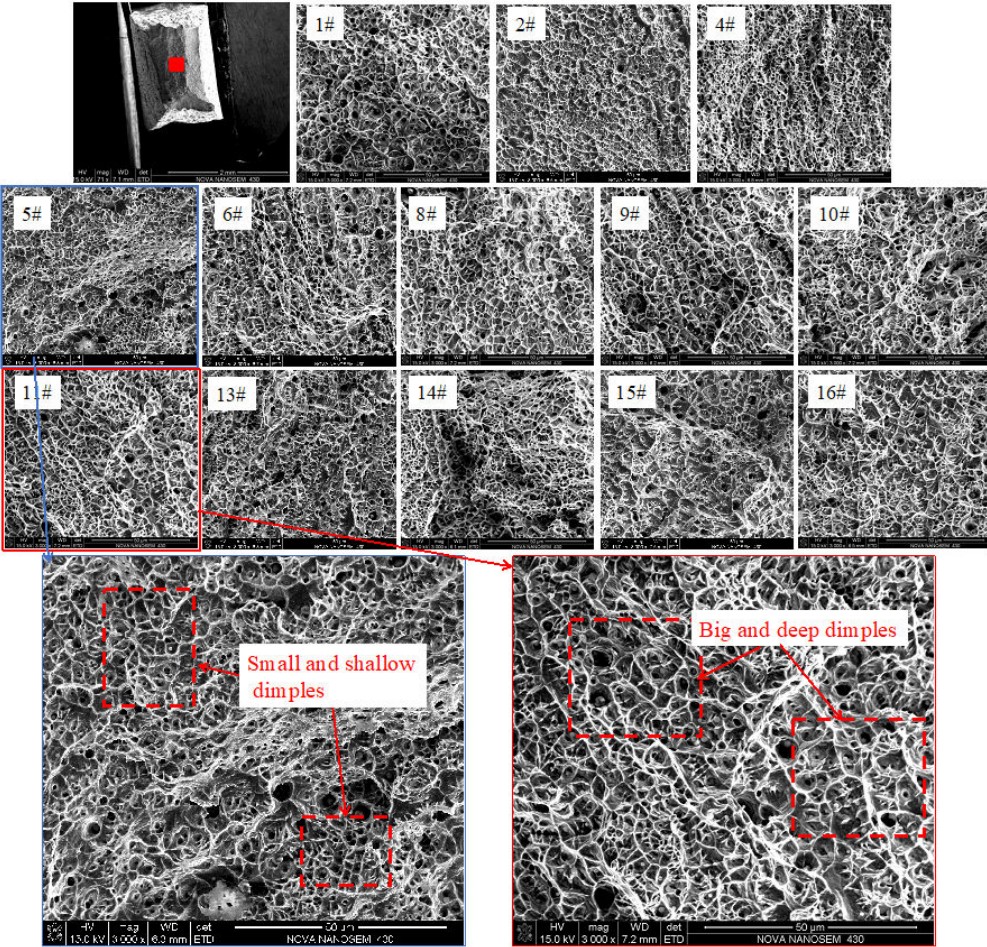

**Figure 7.** SEM images of fracture surface (SEM represents scanning electron microscope).

According to the above analysis, hardness, tensile strength and elongation were selected for evaluating the performance by the evaluation system, and the normalized performance is shown in Table 3. Due to the positive relationship of hardness and tensile strength, the index weight was set to 30%, 30% and 40% to obtain the total performance score, respectively. The final performance score in Table 2 was taken from Table 3.

**Table 3.** Performance evaluation results.

| Samples Performance / Normalized Values | 1# | 2# | 4# | 5# | 6# | 8# | 9# | 10# | 11# | 13# | 14# | 15# | 16# |
|---|---|---|---|---|---|---|---|---|---|---|---|---|---|
| Hardness | 0.42 | 1 | 0.76 | 0.57 | 0.20 | 0.60 | 0.68 | 0.22 | 0.12 | 0.14 | 0.05 | 0 | 0.22 |
| Tensile strength | 0.70 | 1 | 0.94 | 0.78 | 0.48 | 0.87 | 0.81 | 0.57 | 0 | 0.58 | 0.31 | 0.21 | 0.28 |
| Elongation | 0.36 | 0.15 | 0.17 | 0 | 0.62 | 0.01 | 0.66 | 0.56 | 1 | 0.64 | 0.45 | 0.61 | 0.76 |
| Total score | 48 | 66 | 57.8 | 40.5 | 45.2 | 44.5 | 71.1 | 46.1 | 43.6 | 47.2 | 28.8 | 30.7 | 45.4 |

### 3.5. Parameters Modeling and Optimal Prediction of Auxiliary Gas Process

After quantifying and normalizing the data, 50% was chosen as the weight of morphology and performance, respectively, and the scores are presented in Table 2.

Design-Expert software was used to analyze the evaluation results, and then the regression equation was obtained as follows:

$$R = 94.62 - 0.021A - 3.14B - 1.03C + 0.105AB - 0.057AC + 0.276BC + 0.0064A^2 - 0.291C^2 - 0.0035A^2B + 0.0017A^2C + 0.0019AC^2$$

where $A$ indicates the angle between the nozzle and the welding torch, $B$ represents the auxiliary gas flow rate, $C$ expresses the distance from the nozzle to the substrate, and $R$ indicates the comprehensive score of the deposition.

Analysis of Variance (ANOVA) is analyzing the contribution of variation from different sources to the total variation, which can determine the influence of controllable factors on the research results [21]. ANOVA of the regression equation is shown in Table 4, and Model F-value of 693.66 implies the model is significant. There is just a 0.01% chance that an *F*-value could occur due to noise. However, the *F*-value of 1.14 implies the Lack of Fit is not significant relative to the pure error, and there is a 34.60% chance that a Lack of Fit could occur due to noise. Moreover, the model correction and prediction decision coefficient are 0.9979 and 0.9728, respectively, and the difference between them is lower than 0.03, which indicates that the model fits well.

**Table 4.** ANOVA analysis of the regression equation of additive manufacturing (AM) synthesis score (ANOVA represents the Analysis of Variance).

| Source | Sum of Squares | df | Mean Square | *F*-Value | *p*-Value | Significance |
|---|---|---|---|---|---|---|
| Model | 1190.90 | 11 | 108.26 | 693.66 | <0.0001 | Significant |
| $A$ | 62.69 | 1 | 62.69 | 401.69 | <0.0001 | - |
| $B$ | 1.78 | 1 | 1.78 | 11.42 | 0.0197 | - |
| $C$ | 51.05 | 1 | 51.05 | 327.10 | <0.0001 | - |
| $AB$ | 0.0002 | 1 | 0.0002 | 0.0015 | 0.9710 | - |
| $AC$ | 17.01 | 1 | 17.01 | 109.02 | 0.0001 | - |
| $BC$ | 190.23 | 1 | 190.23 | 1218.85 | <0.0001 | - |
| $A^2$ | 508.73 | 1 | 508.73 | 3259.49 | <0.0001 | - |
| $C^2$ | 180.98 | 1 | 180.98 | 1159.55 | <0.0001 | - |
| $A^2B$ | 31.16 | 1 | 31.16 | 199.64 | <0.0001 | - |
| $A^2C$ | 7.01 | 1 | 7.01 | 44.90 | 0.0011 | - |
| $AC^2$ | 1.06 | 1 | 1.06 | 6.79 | 0.0479 | - |
| Residual | 0.7804 | 5 | 0.1561 | - | - | - |
| Lack of fit | 0.1730 | 1 | 0.1730 | 1.14 | 0.3460 | Not significant |
| Pure error | 0.6074 | 4 | 0.1519 | - | - | - |
| Total | 1191.68 | 16 | - | - | - | - |

## 4. Discussion

Figure 8a shows the 3D surface diagram of the relationship between the comprehensive score, the auxiliary gas flow rate and the nozzle distance as the auxiliary nozzle angle is 15°, then Figure 8b presents the 3D surface diagram of the relationship between the score, the auxiliary nozzle angle and distance when the auxiliary gas flow rate is 20 L·min$^{-1}$, and Figure 8c expresses the 3D surface diagram of the relationship between the score, the auxiliary nozzle angle and the gas flow at the distance of 9 mm. Therefore, it can be seen from Figure 8a that when the angle remained at 15° and the nozzle distance was in the range of 9.5 to 14 mm, the score probably had an increasing trend with a rising gas flow rate, while when the nozzle distance was from 4 to 9.5 mm, the score probably had a decreasing trend along with increasing the gas flow rate. When the flow rate was in the range of 15 to 17 L·min$^{-1}$, the score tended to rise with increasing the nozzle distance, while the flow rate was from 17 to 25 L·min$^{-1}$, the score increased first and then decreased with the increase of nozzle distance.

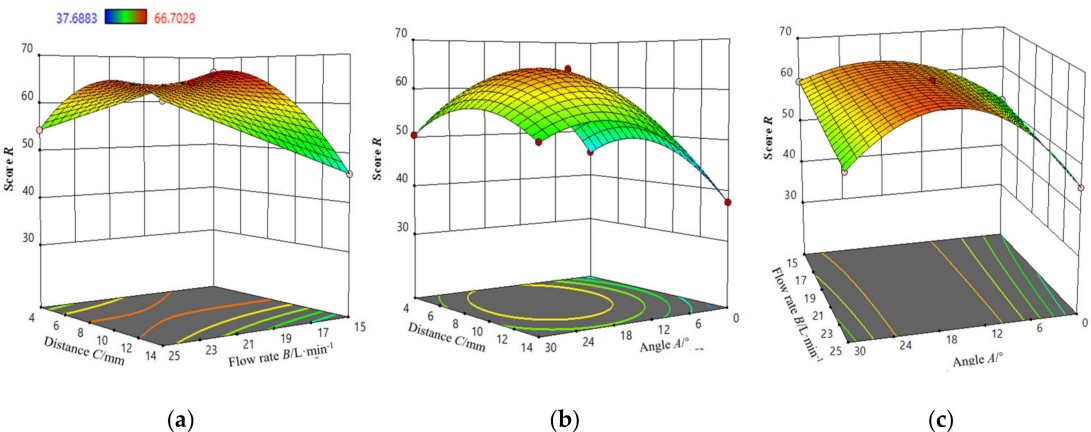

**Figure 8.** Analysis by 3D response surface of auxiliary gas scoring, (**a**) nozzle angle of 15°, (**b**) gas flow of 20 L·min$^{-1}$, (**c**) nozzle distance of 9 mm.

Therefore, based on the above analysis, the curvatures of variables were different and the influence of the auxiliary gas nozzle angle $A$, gas flow rate $B$ and nozzle height $C$ on the comprehensive score of AM samples was not completely independent, and each factor had an interactive effect.

Depending on the range of three parameters, 21 optimal schemes were obtained by the model prediction. Considering that deposition spatter would have an effect on using the nozzle subsequently when the nozzle distance was below 6 mm, and the larger the gas flow rate, the more obvious the stirring effect on the unsolidified molten pool, consequently, 17.4°, 25 L·min$^{-1}$ and 10.44 mm are chosen as the optimal auxiliary gas process parameters, and the total score can reach 68.

## 5. Conclusions

This study proposed a two-direction auxiliary gas process applied in double-wire SS316L stainless steel arc additive manufacturing by CMT + P mode. Based on the BBD response surface method and evaluation system, the regression equation between three auxiliary gas process parameters and the total molding and performance score was established. The conclusions obtained from the results are as follows:

(1) A two-direction auxiliary gas process platform was constructed to explore the influence of auxiliary gas on the liquid pool and solidified deposition in the manufacturing process; that is, the auxiliary gas jet could stir and cool the unsolidified molten pool, which would further improve the morphology, microstructure and the properties.

(2) Based on the image recognition method, the characteristic parameters of the deposited cross-section were extracted to obtain quantitative results, such as molding efficiency and roughness. Then an evaluation system of normalized morphology, hardness and tensile data was built.

(3) An auxiliary gas nozzle angle of 30° produced defects easier but had good properties for strongly stirring and pushing the molten pool forward, while the angle of 0° made the bottom forming and mechanical properties worse than the angle of 30°.

(4) Increasing the auxiliary gas flow to 25 L/min or decreasing distance to 4 mm enhanced the strength and stiffness of the airflow, and then improved the ability to stir the molten pool and resist the interference.

(5) The optimal auxiliary gas process parameters of 17.4°, 25 L/min and 10.44 mm were obtained by the verified model based on the BBD response surface method.

**Author Contributions:** Methodology, W.W., P.Y. and J.X; Writing—Original draft preparation, W.W. and W.X.; project administration and funding acquisition, P.Y., J.X. and W.X.; data curation, W.W. and P.Y.; Writing—Review and editing, providing ideas, W.W., H.L. and H.T. All authors have read and agreed to the published version of the manuscript.

**Funding:** This research was funded by the National Natural Science Foundation Project of China, grant number 51875213 and 51805099; Guangzhou Science and Technology Plan Project, grant number 201805010001; Project of Educational Commission of Guangdong Province of China, grant number 2020ZDZX2019; and Science and Technology Planning Projects of Guangzhou, grant number 201803030041 and 201905010007.

**Conflicts of Interest:** The authors declare no conflict of interest.

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
