# Peer review of "Parameters Optimization of Auxiliary Gas Process for Double-Wire SS316L Stainless Steel Arc Additive Manufacturing"

_metals, doi:10.3390/met11020190_

Round 1

Reviewer 1 Report

The results in this paper are interesting.  However, some parts need to be rewritten to make them clearer to the reader.

The results in this paper are interesting.  However, some parts need to be rewritten to make them clearer to the reader.

Line 26 and following

The objectives of the research are given in the abstract, but not in the introduction which deals with a list of different arc modes and materials experimented in literature.  

Therefore, the choice of a double wire arc system adopted by the authors should be justified in the light of the exposed state of art; then the goals of the work should be clearly given at the end of the introduction. The reader must be able to assess easily whether the conclusions match with the research objectives.

Line 96

What is the material of which the plate is made?

Line 112-118

In my opinion this explanation could confuse the reader.

Line 179

It would be better to separate the results from the discussion (the discussion should start from line 255)

Line 180

Is BBD the acronym of Box-Behnken Design? Please clarify this procedure and add a reference.

Line 181-182

What do you mean for this sentence? The test is repeated only for the sample 1#? These lines need to be written more clearly.

Line 184

Specify that the “performance score” in table 2 is taken from table 3.  It is not usual to anticipate the results that are shown in a next table.

Line 188

In table 2 substitute the “Height between nozzle and substrate” with the “Distance between nozzle and substrate”, as it is defined in line 147.

Line 255

Start the Discussion

Line 265

ANOVA is for Analysis of Variance.  Please spend a few words to describe the method, by indicating the bibliographic reference as well.

Figure 8

In line 263 it is stated that “C expresses the distance from the nozzle to the substrate”.  So, it is better to keep this definition also in fig. 8. Please correct the word “height” with “distance” in the labels on the axes in fig. 8 a) and b) and in the caption of fig. c).

Author Response

Dear Reviewer: Thank you very much for your letter and comments/suggestions concerning our manuscript entitled “Parameters Optimization of Auxiliary Gas Process for Double-wire Arc Additive Manufacturing” (Manuscript ID: metals-1070680). The comments and suggestions are very valuable and helpful for us to improve the quality of the paper. We have carefully considered the comments/suggestions and have made a substantial revision to the manuscript. Revised places are marked by the "Track Changes" function in Microsoft Word in the text of the manuscript. Meanwhile, we have also prepared a list of changes and responses to the review comments as follows: The results in this paper are interesting. However, some parts need to be rewritten to make them clearer to the reader. 1. Line 26 and following The objectives of the research are given in the abstract, but not in the introduction which deals with a list of different arc modes and materials experimented in literature. Therefore, the choice of a double wire arc system adopted by the authors should be justified in the light of the exposed state of art; then the goals of the work should be clearly given at the end of the introduction. The reader must be able to assess easily whether the conclusions match with the research objectives. Response 1: Thank you very much for your kind comments and useful suggestions! We are sorry for accidentally deleting the summary paragraph in the introduction when modified the format. We have added the objectives of the research at the end of the introduction. 2. Line 96 What is the material of which the plate is made? Response 2: The material of the plate in this study is SS316L. 3. Line 112-118 In my opinion this explanation could confuse the reader. Response 3: Thank you for your kind comment! We have rectified the sentence to easy understand for reader in line 147-148. 4. Line 179 It would be better to separate the results from the discussion (the discussion should start from line 255) Response 4: Thanks for your useful suggestion! We have separated the results from the discussion, and the discussion starts from line 336. 5. Line 180 Is BBD the acronym of Box-Behnken Design? Please clarify this procedure and add a reference. Response 5: Yes, BBD is the abbreviation of “Box-Behnken Design”, and we have clarified this procedure in line 117-122 and added two references. 6. Line 181-182 What do you mean for this sentence? The test is repeated only for the sample 1#? These lines need to be written more clearly. Response 6: Thank you for your kind comments and useful suggestions! We have rewritten this sentence clearly. Based on Box-Behnken Design (BBD) response surface method, this study designed 17 samples, of which had 5 repeated samples as 1#, 3#, 7#, 12# and 17#, therefore, sample 1# was the representation in the following studies. 7.Line 184 Specify that the “performance score” in table 2 is taken from table 3. It is not usual to anticipate the results that are shown in a next table. Response 7: Thanks for your useful suggestion! We have specified that the “performance score” in Table 2 was taken from Table 3. 8.Line 188 In table 2 substitute the “Height between nozzle and substrate” with the “Distance between nozzle and substrate”, as it is defined in line 147. Response 8: Thank you for your useful suggestion! We have substituted the “Height between nozzle and substrate” in Table 2 with the “Distance between nozzle and substrate”. 9.Line 255 Start the Discussion Response 9: Thank you for your useful suggestion! We have started the discussion from line 336. 10.Line 265 ANOVA is for Analysis of Variance. Please spend a few words to describe the method, by indicating the bibliographic reference as well. Response 10: Thank you for your kind comments and useful suggestions! We have added a paragraph and reference to describe the method of ANOVA in line 328-330. 11. Figure 8 In line 263 it is stated that “C expresses the distance from the nozzle to the substrate”. So, it is better to keep this definition also in fig. 8. Please correct the word “height” with “distance” in the labels on the axes in fig. 8 a) and b) and in the caption of fig. c). Response 11: Thank you for your useful suggestions! We have corrected the word “height” with “distance” in labels on the axes in Fig.8 a) and b), and caption of Fig.8 c). Finally, we appreciate very much for your time in reviewing our manuscript. Best wishes! Wei Wu, Jiaxiang Xue, Wei Xu, Hongyan Lin, Heqing Tang.

Reviewer 2 Report

This manuscript is well written and of interest to the AM community. I therefore recommend it in its current form. Please double check grammar. Shorten excessively long sentences to improve readability of the article.

Author Response

Dear Reviewer: Thank you very much for reviewing our manuscript entitled “Parameters Optimization of Auxiliary Gas Process for Double-wire Arc Additive Manufacturing” (Manuscript ID: metals-1070680). The comments and suggestions are very valuable and helpful for us to improve the quality of the paper. We have carefully considered the comments/suggestions and have made a substantial revision to the manuscript. Revised places are marked by the "Track Changes" function in Microsoft Word in the text of the manuscript. Meanwhile, we have also prepared a list of changes and responses to the review comments as follows: This manuscript is well written and of interest to the AM community. I therefore recommend it in its current form. Please double check grammar. Shorten excessively long sentences to improve readability of the article. Response: Thank you for your kind comments and useful suggestions! We have checked the grammar and shorten excessively long sentences to improve readability of the article. Best wishes! Wei Wu, Jiaxiang Xue, Wei Xu, Hongyan Lin, Heqing Tang, Ping Yao.

Reviewer 3 Report

Abstract is little bit poor formulated and I suggest rephrasing in in order to ease follow and understand its meanings. Now seems everything is based in “ auxiliary “

Apart the English should eb considerable adjusted

“ It has advantages” as an example …however there are plenty errors everywhere

I am not sure if “ and relatively low production” is an advantage

A better structure introduction is required

There is difficult to understand the industrial impact of this work

Which is you input ???novelty is missing

What is the “the index weight”???

How many sample were verified for each type of trial in order to calculate the average ??

How was prepared the microstructure to be analysed ? please provide details of preparation method

What does means “BBD”??

Figure 4 present many info but difficult to interpret them, I suggest to further elaborate deials and discuss what you have noted for each samples …apart I suggest to put some zoom images in order to see clearly where is the dendrite or other  structure

“of 2# was the largest” replace with “of 2# was the highest”

As per Fig 4 please further elaborate for Fig 5, 6..for example which is the sample “with the auxiliary torch angle of 30°”???

From my first impression from Figure 7 there all the samples indicate ductile fracture..please put some zoom images to show if you achieved other microstructural patterns

There is need a section with discussion cause you not have interpreted the results at all

What is the image recognition method and where did you have used

I suggest to formulate the conclusion per your achievements

Author Response

Dear Reviewer: Thank you very much for reviewing our manuscript entitled “Parameters Optimization of Auxiliary Gas Process for Double-wire Arc Additive Manufacturing” (Manuscript ID: metals-1070680). The comments and suggestions are very valuable and helpful for us to improve the quality of the paper. We have carefully considered the comments/suggestions and have made a substantial revision to the manuscript. Revised places are marked by the "Track Changes" function in Microsoft Word in the text of the manuscript. Meanwhile, we have also prepared a list of changes and responses to the review comments as follows: 1. Abstract is little bit poor formulated and I suggest rephrasing in order to ease follow and understand its meanings. Now seems everything is based in “ auxiliary “ Response 1: Thank you for your kind comments and useful suggestions! We have rephrased the abstract to easy follow and understand its meanings. 2.Apart the English should be considerable adjusted “ It has advantages” as an example …however there are plenty errors everywhere Response 2: Thanks for your kind comments and useful suggestions! English have been considerable adjusted. 3.I am not sure if “ and relatively low production” is an advantage Response 3: Thank you for your careful review! We made a mistake of “relatively low production” and have corrected it with “high production” in line 39. 4.A better structure introduction is required There is difficult to understand the industrial impact of this work Which is you input ???novelty is missing Response 4: Thank you for your kind comments and careful review!! We are sorry for accidentally deleting the summary paragraph in the introduction when modified the format. We have added the objectives of the research at the end of the introduction in line 112-124. 5.What is the “the index weight”??? Response 5: “The index weight” means that the value and relative importance value of each index of a tested object in the whole. Generally, according to the principle of statistics, the sum of the weight of each index is regarded as 1 (that is, 100%), and the weight of each index is expressed as a “the index weight” or "weight coefficient". 6.How many sample were verified for each type of trial in order to calculate the average ?? Response 6: Three samples were verified for each type of trial in order to calculate the average. 7.How was prepared the microstructure to be analysed ? please provide details of preparation method Response 7: Thank you for your kind comments and useful suggestions! We have added the preparation method of microstructure samples in line 206-214. 8.What does means “BBD”?? Response 8: Response surface method is an experimental design and analysis method that uses mathematical and statistical methods to model, analyze and optimize multi factor response problems through reasonably designed experiments, so as to finally achieve the optimization of objectives. Box-Behnken Design (BBD) is a type of response surface design, which does not include embedding factor or partial factor design. BBD has a processing combination located at the midpoint of the edge in the experimental space, and requires at least three factors. 9.Figure 4 present many info but difficult to interpret them, I suggest to further elaborate deials and discuss what you have noted for each samples …apart I suggest to put some zoom images in order to see clearly where is the dendrite or other structure Response 9: Thank you for your kind comments and useful suggestions! We have added the zoom microstructural image of sample #11 to show the dendrite clearly. 10.“of 2# was the largest” replace with “of 2# was the highest” Response 10: Thank you for your useful suggestion! We have replaced “of 2# was the largest” with “of 2# was the highest” in line 273. 11. As per Fig 4 please further elaborate for Fig 5, 6..for example which is the sample “with the auxiliary torch angle of 30°”??? Response 11: Thank you for your kind comments and useful suggestions! We have elaborated the samples for Fig. 5 in line 273-277. 12.From my first impression from Figure 7 there all the samples indicate ductile fracture..please put some zoom images to show if you achieved other microstructural patterns Response 12: Thanks for your useful suggestions! We have provided zoom images of samples #5 and #11, which shows ductile fracture. 13.There is need a section with discussion cause you not have interpreted the results at all Response 13: Thank you for your kind comment and useful suggestion! We have separated the results from the discussion, and the discussion starts from line 336. 14.What is the image recognition method and where did you have used Response 14: Based on LabVIEW software, edge recognition and measurement on the cross-sectional image of depositions were used to obtain the features, such as the maximum height and width, the maximum internal rectangular height and width of the cross-section, etc. The image recognition method was used to calculate the surface roughness and morphology efficiency of deposition in section 2.2. 15.I suggest to formulate the conclusion per your achievements Response 15: Thank you for your kind comments and useful suggestions! We have formulated the conclusion per achievement. Best wishes! Wei Wu, Jiaxiang Xue, Wei Xu, Hongyan Lin, Heqing Tang, Ping Yao.

Round 2

Reviewer 3 Report

.